# Metabolome and Transcriptome Analysis of Hexaploid *Solidago canadensis* Roots Reveals its Invasive Capacity Related to Polyploidy

**DOI:** 10.3390/genes11020187

**Published:** 2020-02-10

**Authors:** Miao Wu, Yimeng Ge, Chanchan Xu, Jianbo Wang

**Affiliations:** College of Life Sciences, Wuhan University, Wuhan 430072, China; wumiao@whu.edu.cn (M.W.); emeng@whu.edu.cn (Y.G.); xuchanchan@whu.edu.cn (C.X.)

**Keywords:** cytotype, plant invasion, transcriptome, metabolome, *Solidago canadensis*, allelopathy

## Abstract

Polyploid plants are more often invasive species than their diploid counterparts. As the invasiveness of a species is often linked to its production of allelopathic compounds, we hypothesize that differences in invasive ability between cytotypes may be due to their different ability to synthesize allelopathic metabolites. We test this using two cytotypes of *Solidago canadensis* as the model and use integrated metabolome and transcriptome data to resolve the question. Metabolome analysis identified 122 metabolites about flavonoids, phenylpropanoids and terpenoids, of which 57 were differentially accumulated between the two cytotypes. Transcriptome analysis showed that many differentially expressed genes (DEGs) were enriched in ‘biosynthesis of secondary metabolites’, ‘plant hormone signal transduction’, and ‘MAPK signaling’, covering most steps of plant allelopathic metabolite synthesis. Importantly, the differentially accumulated flavonoids, phenylpropanoids and terpenoids were closely correlated with related DEGs. Furthermore, 30 miRNAs were found to be negatively associated with putative targets, and they were thought to be involved in target gene expression regulation. These miRNAs probably play a vital role in the regulation of metabolite synthesis in hexaploid *S. canadensis*. The two cytotypes of *S. canadensis* differ in the allelopathic metabolite synthesis and this difference is associated with regulation of expression of a range of genes. These results suggest that changes in gene expression may underlying the increased invasive potential of the polyploidy.

## 1. Introduction

Invasive plants often cause biodiversity loss and serious damage to the ecosystem [1,2]. Understanding how they invade ecosystems is a major challenge for many ecologists. One of the possible explanations of increased species invasiveness may be polyploidization, as polyploid plants are more likely to become invasive than their diploid counterparts [3]. Polyploidy is common in most flowering plants and may facilitate species invasiveness in agricultural systems and natural habitats [4] thanks to extensive changes in genetic make-up and morphological and physiological traits of the plants [5,6]. Some polyploid plants successfully and effectively settle in new habitats in highly fluctuating environments was due to their potential for invasion [7,8]. One of the important traits previously suggested as affecting invasive potential is production of allelochemicals [9], which significantly affect the germination and growth of native species [10]. On the other hand, polyploid plants have the ability to enhance the quantity of metabolites compared with diploid plants, such as *Dendrobium* and *Cymbopogon* [11,12]. 

*Solidago canadensis* L., a perennial plant of the Asteraceae family native to North America, with three cytotypes, diploid (2*n* = 2x =18), tetraploid (2*n* = 4x = 36) and hexaploid (2*n* = 6x = 54) [13]. In the nonnative range, *S. canadensis* is highly abundant and affects the diversity of communities in the invaded ranges [14]. The hexaploid *S. canadensis* can establish a monodominant community in the wild, and this invasion behavior has a strong relationship with the ability to synthesize allelochemical compounds [15], which are able to inhibit the growth of the organisms around them [16,17]. Thus, hexaploid *S. canadensis* may have stronger allelopathy ability than diploid *S. canadensis* in nonnative ranges, which may have an impact on its invasive effects. 

Polyploid plants can change gene expression and alter phenotype, thereby favoring themselves for developing phenotypic innovation [18]. In addition, some miRNAs are major biological factors regulating metabolite synthesis in plants [19]. Therefore, to gain a more complete picture, the metabolome and transcriptome data of two cytotypes of *S. canadensis* roots provide insight into the regulation of gene expression with metabolic networks. We now have a great new tool, ultraperformance liquid chromatography-mass spectrometry (UPLC-MS), which allows to analyze metabolic profiles. It is useful because of its high chromatographic resolution coupled with high mass accuracy, and it can identify putative metabolites [20]. Furthermore, next-generation sequencing (NGS) technology can provide large amounts of data with greater sequencing depth to better understand significant changes in gene expression processes in non-model organisms, which facilitates gene discovery and comparative transcription to identify candidate genes [21,22]. On the other hand, the deep sequencing techniques used to characterize miRNA frequencies and identify new miRNAs further identify the complexity of regulatory networks in plants [23]. Through a combination of qualitative and quantitative analysis of the metabolome with the transcriptome, the major metabolism pathway-related candidate genes and miRNAs were identified in *S. canadensis*. 

In this study, we explored the difference of allelopathic metabolites related genes and miRNAs between hexaploid and diploid *S. canadensis*. Based on this study, we tested the following hypotheses: (i) the production of allelopathic metabolites was different between two cytotypes of *S. canadensis*, indicating that polyploidization is a primary driver of the difference of two cytotypes; (ii) polyploidy can alter the expression of some allelopathic metabolite-related genes and therefore affect the molecular mechanism of allelopathy in plant; and (iii) the increased content of allelopathic compounds in hexaploid cytotype may facilitate its invasiveness. 

## 2. Materials and Methods

### 2.1. Sampling

We transplanted 12 randomly selected rhizomes of hexaploid cytotype (2*n* = 2x = 54) from Wuhan (30°32′N, 114°25′E), Hubei Province, China, and diploid (2*n* = 2x = 18) cytotype from Kunming (24°55′N, 102°47′E), Yunnan Province, China, to the Wuhan University open-air garden. The distances between collected rhizomes in each individual were at least 10 m to reduce the possibility of samples in the same genet. After three months (February to April) of cultivation in the open-air garden, the fresh root tissues from two cytotypes were collected and placed in liquid nitrogen immediately and stored at −80 °C for further investigation. The three replicate samples from each cytotype were taken from individual plant for RNA-seq, and 12 replicate samples from each cytotype were taken from individual plant for HPLC-MS.

### 2.2. Root Metabolite Extraction

Twelve individual samples from each cytotype of *S. canadensis* roots were provided for the metabolome analysis. Frozen samples of 25 mg of root tissue were ground to homogeneity (40 Hz, 5 min) under freezing conditions. Each sample was added to 800 μL of precooled extraction buffer (mixture of methanol and water of equal volume), deposited at −20 °C for 2 h and centrifuged (15 min, 25,000 g) at 4 °C. Finally, 550 μL of supernatant liquid from each sample was transferred to a new EP tube for testing, and 30 μL of each sample was mixed as a quality control (QC) [20].

### 2.3. Ultraperformance LC Quadrupole Time-of-Flight Tandem Mass Spectrometry (UPLC-Q-TOF-MS) Metabolite Profiling and Identification

To compare the metabolite composition involved in the two cytotypes of *S. canadensis*, non-biased metabolite profiling of *S. canadensis* roots was analyzed using HPLC-MS. The order of samples run for metabolomic analyses was 10 QC samples firstly, and then insert a QC sample after each 10 randomly detect samples, finally, three QC sample was run for finish. A 10 μL injection volume of each sample was used in a UPLC system (ACQUITY UPLC; Waters, UK) and a hybrid QTOF tandem mass spectrometer (Waters, UK) for metabolite profiling and identification. All chromatographic separations were performed on an ACQUITY UPLC BEH C18 column (100 mm × 2.1 mm, 1.7 μm, Waters, UK) using mobile phase A (0.1% formic acid in deionized water) and mobile phase B (0.1% formic acid in acetonitrile). The column oven was maintained at 50 °C, and the flow rate was maintained at 0.4 mL min^-1^, and the mobile phase consisted of solvent A (0.1% formic acid). Eluted metabolites were analyzed in both positive [electrospray ionization-positive (ESI^+^)] and negative (ESI^–^) modes. LC-MS data files including MS^1^ (primary level of mass spectrometry) and MS^2^ (secondary level of mass spectrometry) spectra data were further analyzed using Progenesis QI software (version 2.2) with parameter and outputted as a retention time *m/z* dataset. The mass ions of all samples with relative standard deviation (RSD) > 30% were removed, and the putative compound were searched against the Kyoto Encyclopedia of Genes and Genomes (KEGG) database (https://www.kegg.jp/kegg/) based on the acquired ion peaks and MS^1^ and MS^2^ spectral data. All of the metabolome data was submitted into the Metabolomics Workbench database (https://bigd.big.ac.cn/databasecommons/database/id/6924).

### 2.4. Differentially Accumulated Metabolite Analysis

The intensities of mass peak data from 24 samples (two cytotypes × twelve biological replicates) were preprocessed and normalized in metaX software [24] by the probabilistic quotient normalization (PQN) method. The fold change with |log_2_ Ratio| > 1.00 was used for the univariate statistical analysis process with adjusted *p-*value ≤ 0.05 which corrected by FDR. Principal component analysis (PCA) offered a general overview of the variance in metabolites, and partial least-squares discriminant analysis (PLS-DA) was used for multivariate statistical analysis with the parameters of R^2^ > 0.9, Q^2^ > 0.9 and variable importance in projection (VIP ≥ 1.0) [25] to screen differences in metabolic composition between two cytotypes.

### 2.5. KEGG Pathway Analysis for Differentially Accumulated Metabolites

The differentially accumulated metabolites were annotated by the KEGG database and mapped to KEGG pathways. Based on the total putative identified metabolites as background in each pathway, we used a hypergeometric test to find the significantly enriched KEGG pathways for the differentially accumulated metabolites by R software. The *p*-values were adjusted by FDR with a threshold less than or equal to 0.05.

### 2.6. cDNA Library Construction and Sequence Assembly

Total RNA from *S. canadensis* root tissues was isolated using TRIzol reagent and digested with DNase I. RNA integrity was detected by an Agilent 2100 Bioanalyzer (Agilent RNA 6000 Nano Kit) and satisfied with a score above 7.5 for the sequence. A NanoDrop spectrophotometer (Thermo Scientific, CA, USA) was used to quantify RNAs. The mRNAs were collected by magnetic beads with oligo (dT) and then broken into short fragments. Interrupted mRNA was used as a template to construct a paired-end library. All libraries were sequenced on an Illumina HiSeqTM 4000 (Illumina Inc., MI, USA) for at least paired-end sequencing and with the 6 G database for each sample. The clean reads were obtained by filtering and removing the low-quality reads with more than 20% of the base qualities lower than 10, reads with adaptors and reads with unknown bases greater than 5%. Then, the reads were assembled into unigenes by Trinity (v2.0.6) with parameters: --min_contig_length 150 --CPU 8 --min_kmer_cov 5 --min_glue 5 --bfly_opts ’-V 5 --edge-thr = 0.1 --stderr [26] and Tgicl (v2.0.6) software with parameters: -l 40 -c 10 -v 25 -O ’-repeat_stringency 0.95 -minmatch 35 -minscore 35′ [27]. All clean reads were deposited in the NCBI Sequence Read Archive under the BioProject accession PRJNA514984.

### 2.7. Functional Annotation and Transcription Factor Gene Identification

To identify transcripts that were homologous with genes from model species, all transcripts were aligned with the public databases of the KEGG (http://www.genome.jp/kegg) [28] by BLASTX, and Gene Ontology (GO) (http://geneontology.org) [29] using Blast2GO software with default parameter (https://www.blast2go.com) [30]. Furthermore, to clearly show the distribution of genes related to secondary metabolite synthesis pathways, all sequences were mapped in Mercator (http://www.plabipd.de/portal/mercator-sequence-annotation) and analyzed by MapMan software (http://mapman.gabipd.org/home) [31]. As the ORF of each unigene was tested by getorf software (version: EMBOSS: 6.5.7.0) [32] with –minsize 150, the ORF was aligned with a transcription factor (TF) domain using hmmsearch software (version: v3.0) with the default parameters [33]. In addition to the characteristics of the TF gene family from the data of PlntfDB (http://planttfdb.cbi.pku.edu.cn), the unigenes for the transcription factor family were identified.

### 2.8. Calculation of Gene Expression Level and Analysis of Differently Expressed Genes

Clean reads were mapped to unigenes using Bowtie2 (version: v2.2.5) with parameters set as follows -q --phred64 --sensitive --dpad 0 --gbar 99999999 --mp 1,1 --np 1 --score-min L,0,-0.1 -I 1 -X 1000 --no-mixed --no-discordant -p 1 -k 200 [34], and the gene expression level of each sample was calculated by RSEM software (v1.2.12) with default parameters [35] using fragments per kilobase of transcript per million mapped reads (FPKM) analysis with the default parameters. The differential expression of genes was identified with the threshold of |log_2_ Ratio| > 1.00 and adjusted *p*-value < 0.001 which was corrected by FDR for comparisons between diploid and hexaploid cytotypes with three biological replicates [36]. The annotated differentially expressed genes (DEGs) were mapped to GO terms (or KEGG pathways), and gene numbers for every term (or pathway) were calculated. We used a hypergeometric test to find the significantly enriched GO terms or KEGG pathways among DEGs by R software with all of KEGG annotated genes as background. The threshold of corrected *p*-value ≤ 0.05 was adjusted by FDR.

### 2.9. Association Analysis of the Metabolome and Transcriptome Data Sets

To obtain a systematic view of the variations in metabolites and their corresponding genes, Pearson correlation coefficients were calculated using the R program for metabolome and transcriptome data association. According to the result of KEGG enrichment analysis for the metabolome and transcriptome, the normalized metabolite quantity of phenylpropanoids, flavonoids and terpenoids and expressed genes were calculated. In addition, to model the synthetic and regulatory characteristics of metabolites and related genes, subnetworks were constructed to determine transcript-metabolite correlations. Only the correlation pairs with a Pearson correlation coefficient > 0.99 and *p*-value ≤ 0.05 were selected for further study. The relationships between the metabolome and transcriptome were visualized using Cytoscape (v2.8.2).

### 2.10. Small RNA Library Construction and Sequencing

Total RNAs were separated, and different segment size RNAs, from 18 to 30 nt, were collected by PAGE. The 3′ end of each RNA was linked with a 5′-adenylated and 3′-blocked single-stranded DNA adapter. The RT primer was hybridized to the 3′ adapter, and the 5′ adapter was linked to the 5′ end of the product above. The cDNA library was synthesized and enriched by PCR amplification with the following program: 95 °C for 3 min, 18 cycles of 98 °C for 20 s, 56 °C for 15 s, 72 °C for 15 s. Finally, PCR products of 100~120 bp were separated by PAGE to eliminate primer-dimer products. The Illumina data of small RNAs that contained 5′-adapter contaminants, poly (A) sequences, low-quality reads, sequences lacking the 3′-adapter and sequences without the inset tags were filtered out. The data of small RNA clean reads have been deposited in the NCBI Sequence Read Archive under the BioProject accession PRJNA514984.

### 2.11. Alignment and Identification of Known and Novel miRNA Predication

Clean reads were mapped to the reference genome and identified by public sRNA databases, except that Rfam was aligned by cmsearch software [37] with default parameter. The miRNA precursors were identified by their hairpin structure, which can be used to predict novel miRNAs. In this study, RIPmiR with defult parameter was used to predict novel miRNAs by exploring the characteristic hairpin structure of miRNA precursors [38].

### 2.12. miRNA Expression Analysis and Target Gene Prediction

The transcripts per million (TPM) method was used to calculate the expression of miRNAs. The differential expression of small RNAs was calculated by DEGseq [39], which satisfied |log_2_ Ratio| > 1 and adjusted *p*-value ≤ 0.05. Finding potential target genes for a miRNA is a necessary process for subsequent analysis. To be more accurate, psRobot [40] with parameters as -gl 17 -p 8 -gn 1 and TargetFinder [41] with parameters as -c 4 were used to predict target genes and intersection targets.

### 2.13. RT-qPCR of mRNAs and Stem-Loop RT-PCR of miRNAs

The qRT-PCR primer sequences were designed with Primer 5 software. We performed quantitative PCR on cDNA products with the Step One Real-Time PCR System using the SYBR Green Master Mix (Applied Biosystems) to detect transcript abundance. The miRNA-specific stem-loop primers were designed according to Chen et al. (2005) [42]. The 18S rRNA gene was used as an internal control gene. The experimental reaction conditions were as follows: denaturation at 95 °C 30 s, followed by 40 cycles of denaturation at 95 °C for 15 s, annealing and extension at 60 °C for 1 min. The relative expression levels of selected genes were calculated by the 2^−△△Ct^ method. Each qRT-PCR experiment for mRNAs and miRNAs was carried out in a Step-One Real-Time PCR system with three biological replicates and three technical replicates under the same conditions.

## 3. Results

### 3.1. Putative Metabolite Identification in Roots of Two Cytotypes of S. canadensis

The data from UPLC-Q-TOF-MS were analyzed by principal component analysis (PCA), and the results showed that the two cytotypes clearly separated along the PC1, based on the cytotype tissue specificity (Figure 1). We removed the mass ions with relative standard deviation (RSD) > 30% and acquired 15800 and 6658 mass ions in ESI^+^ and ESI^–^ modes in the roots of *S. canadensis*, respectively. A total of 1305 metabolites were identified (Appendix A) and annotated by the KEGG database in two cytotypes of *S. canadensis*, and assigned to 95 KEGG pathways. Finally, 122 metabolites were identified and classified into three allelopathic compound related groups: terpenoids, phenylpropanoids and flavonoids.

### 3.2. Differentially Accumulated Metabolites between Two Cytotypes of S. canadensis

The score plots (Appendix A) and calculated VIP values from PLS-DA were used to model the differences between the two cytotypes. From 1305 identified metabolites, there were 395 metabolites differentially accumulated between two cytotypes, with 238 increased and 157 decreased in the hexaploid compared to the diploid (Appendix A), based on the VIP (VIP ≥ 1) from PLS-DA combined with univariate statistics |log_2_ Ratio| > 1.00 and adjusted *p*-value ≤ 0.05. From the most of differentially accumulated secondary metabolites, 57 metabolites were classified into three groups, i.e., terpenoids, with 16 more accumulated than hexaploid and 12 less accumulated than diploid, phenylpropanoids, with 7 more accumulated and 3 less accumulated, and flavonoids, with 13 more accumulated and 6 less accumulated (Table 1).

### 3.3. KEGG Enrichment Analysis of Differentially Accumulated Metabolites

KEGG enrichment analysis for differently accumulated metabolites were assigned into 69 KEGG pathways. Our study indicated that 80 and 74 metabolites were assigned to ‘metabolic pathways’ and ‘biosynthesis of secondary metabolites’, respectively. Fifteen metabolites were assigned to ‘diterpenoid biosynthesis’, and ten metabolites were assigned to ‘phenylpropanoid biosynthesis’. Most of the secondary metabolites was significantly enriched in the secondary metabolism synthesis-related pathways, such as ‘flavone and flavonol biosynthesis’, ‘flavonoid biosynthesis’ and ‘sesquiterpenoid and triterpenoid biosynthesis’ (Figure 2A). The metabolites of terpenoids, flavonoids and phenylpropanoids accumulated much more in hexaploid than diploid *S. canadensis*. These secondary metabolites have the allelopathic effect in the invasion process.

### 3.4. Overview of Transcript Construction and Annotation for Two Cytotypes of S. canadensis

A total of 50,888 unigenes were annotated by GO and KEGG databases, with 45,205 in diploid and 41,772 in hexaploid plants. We discovered that 48,340 unigenes were annotated by the KEGG database and classified into some major categories such as ‘metabolism’, ‘genetic information processing’ and ‘environmental information processing’. On the other hand, 19,313 unigenes were assigned to 53 functional groups by GO annotation and primarily classified into three categories: biological process, cellular component and molecular functions. Within the three categories of the GO classification scheme, the dominant terms were ‘catalytic activity’, ‘cellular process’, and ‘cell’. The co-expressed 36,089 unigenes between two cytotypes were detected, and 9116 unigenes were specifically expressed in diploid *S. canadensis*, while 5683 unigenes were specifically expressed in hexaploid *S. canadensis*.

### 3.5. Comparison of Transcriptomes between Two Cytotypes of S. canadensis

We collected 10,565 differentially expressed unigenes between two cytotypes (Appendix A). According to the analysis of GO term enrichment, 4082 DEGs were assigned to the ‘biological process’ category, 5838 DEGs to ‘cellular component’ and 3589 DEGs to ‘molecular function’. 

For the KEGG pathway enrichment analysis, 8943 DEGs were assigned to 135 KEGG pathways in total. The pathway with the most DEGs was ‘metabolic pathways’, followed by ‘biosynthesis of secondary metabolites’, ‘plant-pathogen interaction’ (Appendix A). According to the corrected *p*-value for the pathway enrichment analysis, most DEGs were significantly enriched in allelopathic compound synthesis-related pathways, such as ‘monoterpenoid biosynthesis’, ‘sesquiterpenoid and triterpenoid biosynthesis’, and ‘flavonoid biosynthesis’ (Figure 2B). The GO terms and KEGG pathway enrichment indicated that DEGs related to secondary metabolic processes may play important functions, providing clues for allelopathy studies in hexaploid *S. canadensis*.

### 3.6. Allelopathic Compound Synthesis-Related DEGs

There were some key genes in the secondary metabolite synthesis-related pathways that control the synthesis of allelopathy related compounds. In this study, KEGG analysis identified 38 candidate genes across seven gene families from the *S. canadensis* transcriptome that were associated with the phenylpropanoid pathway (Figure 3A). For example, one *PAL* (phenylalanine ammonia-lyase gene) and three *4CLs* (4-coumarate coenzyme A ligase gene) were annotated. The enzymes encoded by these genes promote the formation of cinnamoyl-CoA and directly affect the terpenoid and flavonoid metabolic pathways. We obtained 76 candidate genes across 26 gene families were related to terpenoid pathways, such as ‘monoterpenoid’, ‘diterpenoid’ and ‘sesquiterpenoid and triterpenoid biosynthesis’ (Figure 3B,C,E). Furthermore, 52 candidate genes across 15 gene families were associated with the flavonoid pathway (Figure 3D), including the gene *FLS*, *F3H* and *DFR*. There were eight *HCTs* encode enzymes that catalyze p-coumaroyl-CoA to form caffeoyl-CoA, which can be further catalyzed by three *CCoAMTs* encoded enzymes to form feruloyl CoA. Five *F3Hs* (flavanone 3-hydroxylase genes) and one *F3’H* (flavonoid 3’-hydroxylase gene) that can encode enzymes catalyze the synthesis of dihydrokaempferol and eriodictyol were also detected. Furthermore, we identified two *DFRs* (dihydroflavonol 4-reductase genes), seven *FLSs* (flavonol synthase genes), and three *ANRs* (anthocyanidin reductase genes) that may encode enzymes to produce numerous flavonoids.

Most of these DEGs that are significantly enriched in terpenoid, flavonoid and phenylpropanoid biosynthesis pathways (Figure 2B) were up-regulated in hexaploid *S. canadensis*. According to MapMan analysis, many genes were also mapped to secondary metabolism-related pathways, such as ‘terpenoids’, ‘phenylpropanoids’, ‘flavonoids’ and ‘phenols’ (Figure 4).

### 3.7. The Regulation of Allelopathic Compound Synthesis-Related Genes between Two Cytotypes

In this study, 454 of 3191 transcription factor genes showed significantly different expression levels in hexaploid *S. canadensis*. The top five differentially expressed TF gene families were *MYB*, *AP2-EREBP*, *WRKY*, *bHLH* and *NAC*, suggesting a key role for these genes in hexaploid *S. canadensis*. Most of these TF genes were up-regulated in hexaploid compared with diploid. Interestingly, some of the differentially expressed TF genes, such as *ERF*, *WRKY33*, *WRKY22* and *WRKY29*, were enriched in the ‘plant-pathogen interaction’ pathway. Some genes that constructed a regulatory network in this pathway, such as *CDPK*, *BAK1*, *MEKK1* (Figure 5A), *PBS1*, *SGT1*, and *RRS1-R* (Figure 5B), were also differentially expressed and mostly up-regulated in hexaploid *S. canadensis*.

Many genes involved in abscisic acid (ABA), brassinosteroid (BR), jasmonic acid salicylic acid and gibberellin (GA) signaling pathways were up-regulated in hexaploid *S. canadensis*. Furthermore, many DEGs that participate in the ‘MAPK signaling pathway’, such as *MEKK1*, *MPK3*, *MPK6 and MPK1/2*, were up-regulated in hexaploid *S. canadensis* compared with diploid *S. canadensis* (Figure 6). These results reveal that the important roles for these DEGs most likely formed regulatory networks for allelopathic compound synthesis; additionally, all candidate genes may provide important clues for further study of allelopathic effects in hexaploid *S. canadensis*.

### 3.8. Correlation Analysis between the Differentially Accumulated Allelopathic Metabolites and Related DEGs

To investigate the association between allelopathic metabolites and related genes in hexaploid *S. canadensis*, the terpenoid-, flavonoid- and phenylpropanoid-related transcripts and their correlations with metabolite abundance were analyzed. According to the KEGG pathway enrichment analysis for the transcriptome and metabolome, we carried out a correlation test for allelopathic metabolite synthesis related DEGs with corresponding metabolite. The results showed that 131 DEGs strongly correlated (R^2^ > 0.99) with 57 metabolites (Appendix A). The interaction network of differentially accumulated secondary metabolites and DEGs was constructed (Figure 7), and some of the correlated DEGs participated in regulating metabolites. As examples, farnesol dehydrogenase gene (CL2850.Contig16_All, CL15067.Contig1_All) can encode enzymes involved in farnesol synthesis, *C3’H* (coumaroylquinate (coumaroylshikimate) 3’-monooxygenase) gene (Unigene7713_All) can encode coumaroylquinate (coumaroylshikimate) 3’-monooxygenase involved in caffeoyl quinic acid synthesis, furthermore, *HST* gene (CL10425.Contig1_All) can encode shikimate O-hydroxycinnamoyltransferase which was affected by caffeoyl quinic acid in the metabolism pathway. The result of correlation analysis suggested that these metabolites may be regulated by related genes, besides, considering that *S. canadnesis* may exist some unique ways of network regulation, we need to provide some novel information for network analysis, and these results require further valid research. By combining transcriptome and metabolome correlation analysis, our data established a comprehensive view of the allelopathic metabolite synthesis regulation network in *S. canadensis* roots. 

### 3.9. Small RNA Expression and Target Gene Prediction

We perfectly matched 54 conserved miRNAs to 21 known miRNA families, with 53 and 54 conserved miRNAs in diploid and hexaploid plants, respectively. Except for the conserved miRNAs, 140 novel miRNAs, with 136 in the diploid plant and 135 in the hexaploid plant were predicted (Table 2). The most abundant known miRNA was the miR166 family. Other miRNA families, such as miR396, miR398, and miR168 families, were also highly expressed.

The number of target genes in both cytotypes of *S. canadensis* were presented in Table 2. These targets contained TF genes that may be involved in plant secondary metabolism, such as *MYB*, *ERF*, and *WRKY*. miR172 was predicted to target sites in the *ERF* gene family of transcription factors, and miR156 targeted the *WRKY* gene. miR166 was predicted to target the *HD-ZIP* TF gene. Other targets, such as zinc finger protein gene (targeted by miR156), *CYP450* gene (targeted by miR167) and *4CL* gene (targeted by novel_mir109, novel_mir104, and novel_mir57), were also considered to be associated with secondary metabolism. These results are consistent with previous research in other plants and suggest that the expression of mRNAs is likely regulated by the coordinated actions of multiple miRNAs in *S. canadensis*.

### 3.10. Visualization of Differentially Expressed miRNA and Target Gene Interaction Network

We identified 76 differentially expressed miRNAs, with 28 up-regulated and 48 down-regulated, in the hexaploid. According to the negative regulatory effect of differentially expressed miRNAs, 90 differentially expressed target genes were collected and formed a connection network (Figure 8). For example, down-regulated miR172a_3 targeted many genes, such as the *ERF* gene (CL4453.Contig3), zinc finger protein gene (CL577.Contig11_All and CL16418.Contig11_All) and *CYP450* monooxygenase gene (CL3396.Contig1_All), which were significantly up-regulated. Down-regulated miR156f participated in the regulation of the RanBP2-type zinc finger protein gene (CL2659.Contig1_All), which was correspondingly up-regulated. Down-regulated miR167d_1 was predicted to regulate the auxin response factor gene (CL1402.Contig5_All), which was also up-regulated in hexaploid *S. canadensis*. Furthermore, miR166a_3p targeted the homeobox-leucine zipper gene (CL2367.Contig4_All), and novel_mir114, novel_mir61, novel_mir39 and novel_mir55 were predicted to target many types of zinc finger protein genes (CL14660.Contig1_All, CL439.Contig1_All, CL439.Contig4_All and CL298.Contig4_All, respectively). These target zinc finger protein genes were all up-regulated in the hexaploid plants. Up-regulated miR845 was predicted to target down-regulated zinc finger protein genes (CL7357.Contig3_All, CL7357.Contig8_All). Our results indicate a potential role for these differentially expressed miRNAs in regulating corresponding genes that regulate secondary metabolism in hexaploid *S. canadensis*.

### 3.11. qRT-PCR and Stem-Loop RT-PCR Confirmed Differentially Expressed mRNAs and miRNAs

To validate the accuracy of transcriptome sequencing for mRNA and miRNA, we performed qRT-PCR and stem-loop RT-PCR on 12 randomly selected DEGs (Appendix A) and nine differentially expressed miRNAs (Appendix A). The trends of relative expression levels determined by qRT-PCR were consistent with RNA-seq. The primer sequences used in this study are listed in Appendix A.

## 4. Discussion

Polyploids are known to have higher invasive potential compared to diploids [3]. However, the molecular mechanism examples that contribute to invasion are lacking. The physiological functions or gene expression may be changed by polyploidy in higher plant species [43]. To provide an extensive characterization of the difference of invasive ability between two cytotypes, we integrated metabolome and transcriptome analysis of the root tissue of *S. canadensis* and showed that the quantities of allelopathic metabolites were significantly different between diploids and hexaploids. In addition, the transcriptome analysis showed that the allelopathic metabolites synthesis-related genes and miRNAs were also differentially expressed between two cytotypes. Our results lay a foundation to understand that polyploidy promotes the invasive ability of hexaploid *S. canadensis* from molecular mechanism.

### 4.1. Allelopathic Compound Synthesis Ability Was Enhanced in Hexaploid S. canadensis

Many allelopathic compounds play an essential role in both of allelopathy and competition process [44]. Polyploidy typically increase metabolic activity through transcriptional divergence, ultimately resulting in changes in metabolite content levels [45]. Many studies have provided evidence that polyploidy promote the production of specific secondary metabolites [46,47]. As an invasive plant, *S. canadensis* has enhanced competitive ability by producing allelopathic compounds [48], which are synthesized by the root of *S. canadensis*, such as flavonoids, phenols and terpenoids [17,49]. In this study, we identified a different quantity of secondary metabolites between two cytotypes of *S. canadensis*. Most of these differentially accumulated secondary metabolites, including terpenoids, phenols and flavonoids, were acted as allelopathic compounds. For instance, germacrene D acts as a sesquiterpenoid responsible for the phytotoxic [50]. Formononetin and myricetin act as flavonoid metabolites that show allelopathic inhibition in many plants [51]. Myrcene is a monoterpenoid metabolite that exists in essential oils in *Salvia* and has an allelopathic effect on radish and garden cress germination [52]. These compounds were accumulated at higher levels in hexaploids (Figure 2A). This result suggests that polyploidy may promote hexaploid *S. canadensis* to possess a greater advantage in allelochemical metabolite synthesis than diploid *S. canadensis*. This may facilitate hexaploid *S. canadensis* to become an invasive species.

### 4.2. Polyploidy Affects the Expression of Allelopathic Metabolite Synthesis Related Genes in Hexaploid S. canadensis

Polyploidy can cause large-scale changes in gene expression levels, and these changes may occur immediately or after several generations, therefore, many of the functional alterations in polyploid plants are correlated with gene expression changes at the transcriptional and posttranscriptional levels [53]. As a result of polyploidy, gene expression levels will be altered [54]. Previous studies have shown that polyploid plants have the ability to synthesize more metabolites than diploid plants [12,47], and related genes have higher expression levels [55]. In this study, many allelopathic metabolites biosynthesis-related DEGs, which were assigned to flavonoid, terpenoid and phenol synthesis-related pathways, were mostly up-regulated in hexaploids. Furthermore, based on the network analysis (Figure 7), most of the differentially accumulated metabolites that possessed allelopathic properties may be regulated by related DEGs. These candidate genes play a key role in phenylpropanoid, terpenoid and flavonoid synthesis. For example, the *PAL* (CL3911.Contig4_All) encodes a key enzyme in the phenylpropanoid pathway responsible for most of the biosynthesis of many secondary metabolites [56]. The *4CL* (CL6667.Contig3_All) gene encodes an essential enzyme responsible for phenylpropanoid and flavonoid synthesis [57]. The *FLS* (CL2039.Contig2_All, CL1610.Contig3_All, CL2205.Contig4_All, CL2205.Contig2_All) gene encodes an enzyme that catalyzes the formation of flavonols [58]. The expression of these DEGs was higher in hexaploids *S. canadensis*. The changed expression of these candidate genes may due to polyploidy. This mechanism may increase production of allelochemicals in hexaploids and thus increasing their invasion potential. 

### 4.3. The Expression of Genes Involved in Regulation May Be Changed in Hexaploid S. canadensis

TFs as regulatory effectors often persist and contribute to gene expression levels in polyploid plants [54]. A large number of TFs can involve plant development and secondary metabolites, such as TIFY TFs play important roles in plant development and hormone response [59], AP2 TFs involve in root development [60], MYB TFs involve in regulating flavonoid biosynthesis under abiotic stress [61]. In this study, many TF genes, such as the *MYB*, *AP2/EREBP*, *bHLH*, and *WRKY* were differentially expressed in hexaploid *S. canadensis* when compared with diploids. These TF genes likely play a crucial role in the regulation of root development and secondary metabolic processes in hexaploid *S. canadensis*, and the specific functions of the above TF genes in *S. canadensis* need further exploration. Plant hormones contribute to plant growth and development and some secondary metabolism-related gene regulation; for instance, jasmonic acid regulates terpenoid and flavonoid biosynthesis in plants [62], GA play multiple roles in plant growth and development processes [63], ABA modulates the growth of primary and lateral roots in Arabidopsis [64,65] and flavonoid synthesis in tomato [66]. Furthermore, MAPK signal transduction can combine with hormone signal response processes and develop into regulation network in root [67,68,69]. These regulation networks which combined with TFs, hormones and MAPK signaling related genes may finally affect secondary metabolite synthesis and favor allelopathic effect in hexaploid *S. canadensis*, thus facilitating this plant becoming an invasive species.

### 4.4. The Post-Transcriptional Regulation May Affect the Gene Expression of Hexaploid S. canadensis

Existing miRNAs are known to be strongly associated with the expression regulation of target genes in plant development [70]. In this study, some differentially expressed miRNAs were predicted to target TF genes, such as *AP2/ERF* and *WRKY*, which are likely related to secondary biosynthesis. For instance, *S. canadensis* miR172a_3 was detected to target *RAP2* (CL2666.Contig13_All), *ERF* (CL4453.Contig4_All, CL4453.Contig5_All, CL2666.Contig5_All, CL4453.Contig8_All), a zinc finger protein gene (CL16418.Contig11_All) and *CYP450* (CL3396.Contig1_All). Some of these target genes are involved not only in the development and hormone signal response of plants but also in the regulation of plant secondary metabolism [71,72,73]. Therefore, compared with diploid, down-regulated expression of miR172a_3 in hexaploid *S. canadensis* may indirectly result in a weak inhibition effect to *ERF* and *CYP450* gene expression, which probably enhanced the expression of these target genes. In addition, miR166 is involved in the targeted regulation of the *HD-zip* gene [74], which participates in the growth of plant roots [75]. Both miR166a-3p and miR166m_2 were predicted to target the *HD-zip* gene (CL2367.Contig4_All), and both of them were down-regulated in the hexaploids compared to the diploids. Correspondingly, the *HD-zip* genes were up-regulated. These miRNAs may be correlated with the growth of plant root tissue and correspondingly altered secondary metabolite synthesis in hexaploid roots. Therefore, miRNAs may indirectly regulate secondary metabolite synthesis in hexaploid *S. canadensis*. 

## 5. Conclusions

This study used metabolome and transcriptome data to analyze differences in metabolite accumulation and gene expression between hexaploid and diploid roots of *S. canadensis*. The results showed a high accumulation of allelochemical metabolites for flavonoids, phenylpropanoids and terpenoids in hexaploid compared with diploid *S. canadensis* roots and identified several candidate genes and miRNAs that might be involved in promoting and regulating allelopathic metabolite synthesis processes. Based on the altered molecular character in hexaploid cytotype, we may understand that polyploidy has promoted hexaploid *S. canadensis* to develop into invasive species. 

## Figures and Tables

**Figure 1 genes-11-00187-f001:**
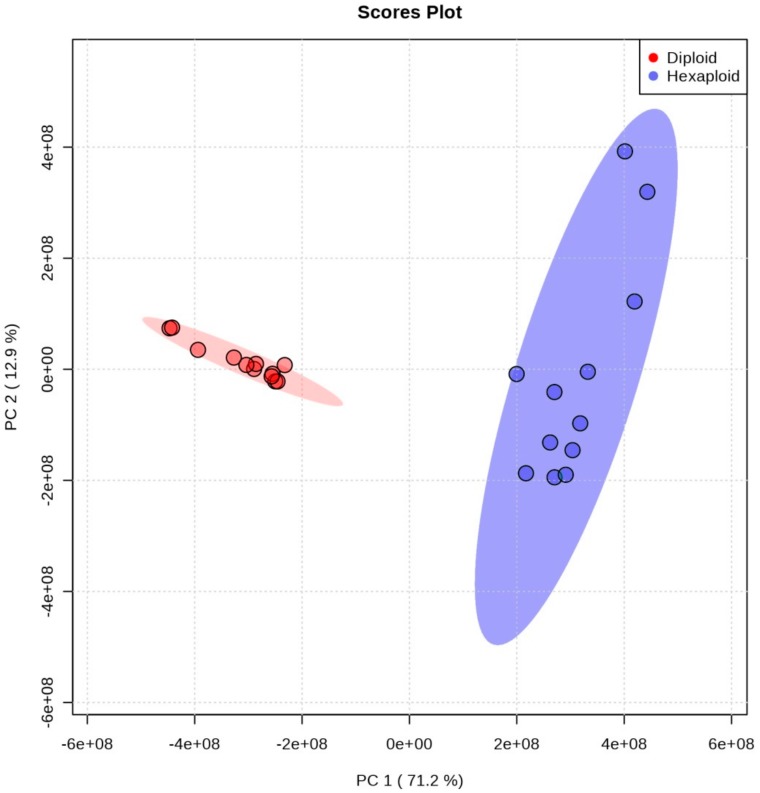
The PCA analysis of metabolites between two cytotypes of *S. canadensis*.

**Figure 2 genes-11-00187-f002:**
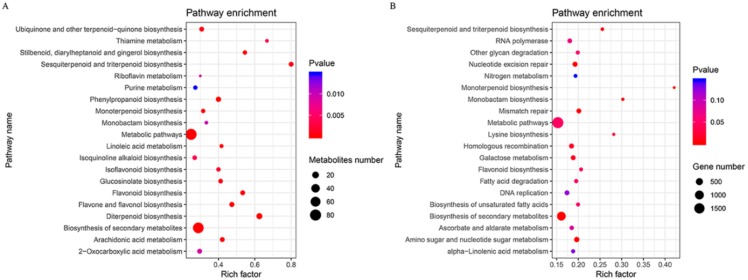
KEGG pathway enrichment analysis. Pathway enrichment of differentially accumulated metabolites (**A**) and differentially expressed genes (**B**). *y*-Axis indicates the pathway name, *x*-axis indicates the enriched factor in each of pathways. The bubble size indicates the number of metabolites or genes. The color bar indicates the corrected p-value, the blue represents higher value, the red represents lower value.

**Figure 3 genes-11-00187-f003:**
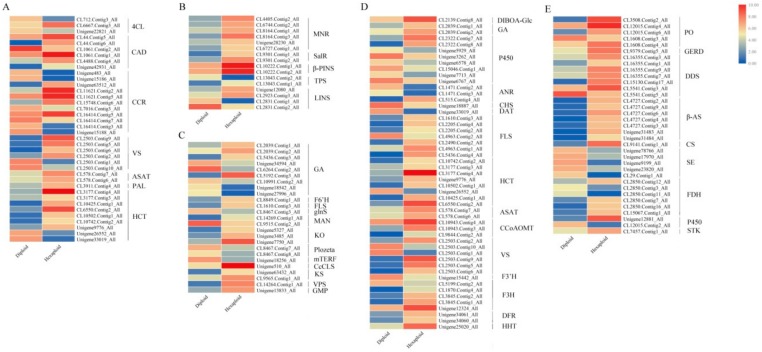
Expression profiles of enriched secondary metabolism synthesis related DEGs. The DEGs related to phenylpropanoid biosynthesis (**A**), monoterpenoid biosynthesis (**B**), diterpenoid biosynthesis (**C**), flavonoids biosynthesis (**D**), and sesquiterpenoid and triterpenoid biosynthesis (**E**) were depected. The color bar indicates the normalized relative expression levels; red indicates high expression level, blue indicates low expression level.

**Figure 4 genes-11-00187-f004:**
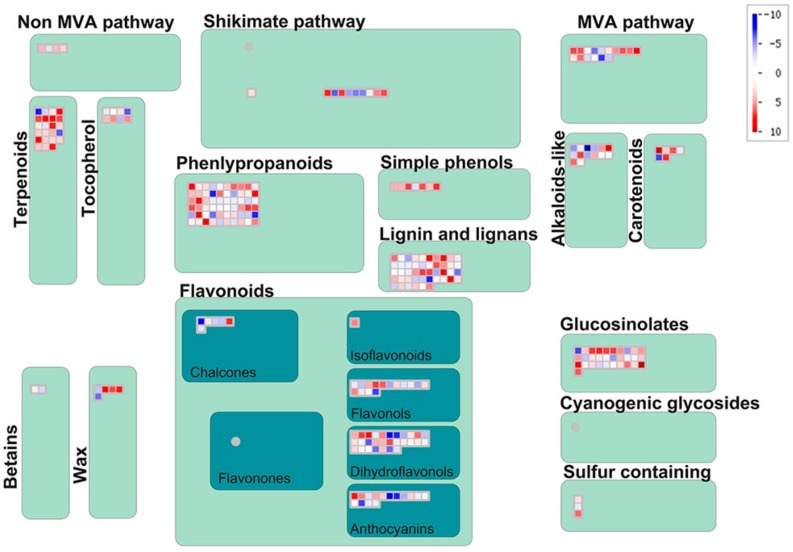
The expression profiles for secondary metabolic-related DEGs analyzed by MapMan. The color bar indicates the normalized log_2_ transformed fold change value in hexaploid compared with diploid, red indicates up-regulation and blue indicates down-regulation.

**Figure 5 genes-11-00187-f005:**
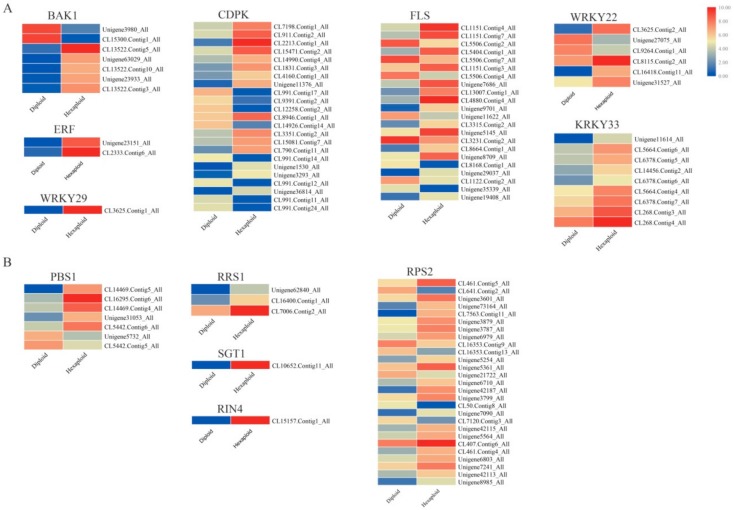
Expression profiles of DEGs related with plant-pathogen interaction pathway. Two major approaches include PAMP-triggered immunity (PTI) (**A**) and effector-triggered immunity (ETI) (**B**). The color bar indicates the normalized relative expression levels; red indicates high expression level, blue indicates low expression level.

**Figure 6 genes-11-00187-f006:**
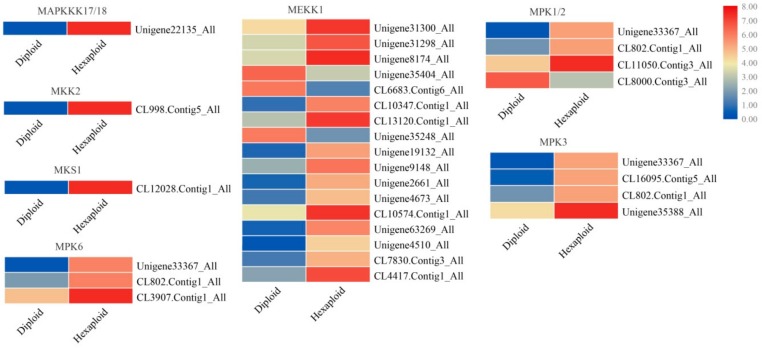
Expression profiles of enriched MAPK signal transduction related DEGs. The DEGs that play a major role in the transduction are depicted. The color bar indicates the normalized relative expression levels; red indicates high expression level, blue indicates low expression level.

**Figure 7 genes-11-00187-f007:**
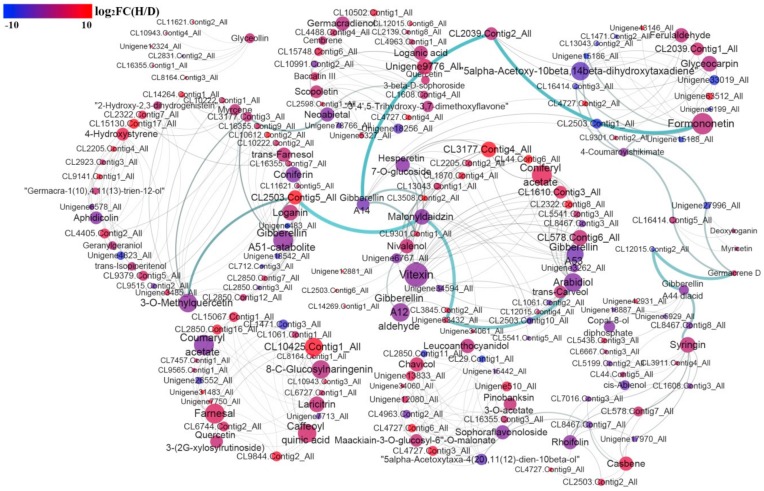
The regulation of DEGs involved differentially accumulated metabolites synthesis network. The color bar represents the log_2_ transformed fold change value of hexaploid compared with diploid. The red represents higher fold change value, and blue represent lower fold change value. The bubble size that based on the connect degree and the width of connect lines that based on edge betweenness was represent the important of nodes for DEGs or differentially accumulated metabolites. Greater bubble size or connecting line thickness represent more important nodes.

**Figure 8 genes-11-00187-f008:**
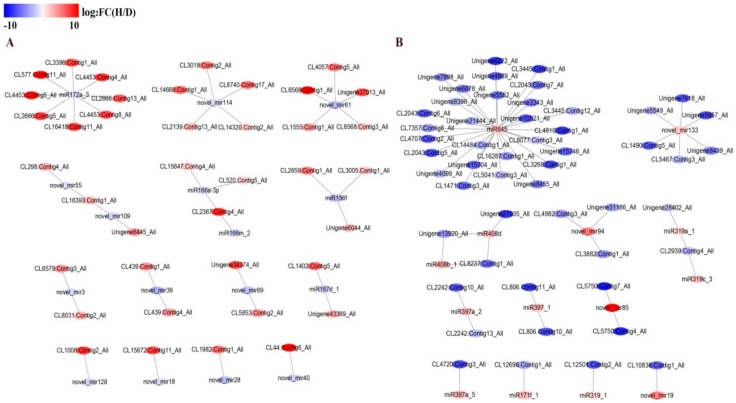
The interaction network of miRNAs with target genes. The down-regulation of miRNAs (**A**) and up-regulation of miRNAs (**B**) negatively correlated with target genes. The color bar indicates the normalized log_2_ transformed fold change value between two cytotypes, red indicates up-regulation and blue indicates down-regulation. The color bar indicates the normalized log_2_ transformed fold change value of hexaploid compared with diploid. The red indicates up-regulation and the blue indicates down-regulation in hexaploid compared with diploid.

**Table 1 genes-11-00187-t001:** Differentially accumulated secondary metabolites between two cytotypes *S. canadensis* root.

Classification	Metabolite Name	log_2_ Fold Change (Hexaploid/Diploid)	Padj-Value
Terpenoid	Casbene	5.275391063	6.49 × 10^−04^
	Farnesal	4.154837827	1.17 × 10^−03^
	Germacrene D	4.08702165	4.16 × 10^−04^
	Baccatin III	3.978134672	6.40 × 10^−05^
	Nivalenol	2.742280848	1.98 × 10^−08^
	Cembrene	2.610800783	3.32 × 10^−03^
	Germacra-1(10),4,11(13)-trien-12-ol	2.563360372	5.93 × 10^−05^
	trans-Carveol	2.255799893	1.33 × 10^−02^
	trans-Farnesol	2.236407545	1.28 × 10^−11^
	trans-Isopiperitenol	2.224502422	1.89 × 10^−03^
	Myrcene	2.199510457	4.49 × 10^−09^
	Loganic acid	2.107521873	3.51 × 10^−06^
	Loganin	2.097243456	6.65 × 10^−06^
	Geranylgeraniol	2.00541636	4.87 × 10^−05^
	Deoxyloganin	1.721160952	3.27 × 10^−07^
	Germacradienol	1.442421048	1.34 × 10^−07^
	Gibberellin A12 aldehyde	−1.218344661	1.23 × 10^−10^
	Gibberellin A51-catabolite	−1.257611873	5.60 × 10^−05^
	Gibberellin A44 diacid	−1.707646412	1.44 × 10^−10^
	Neoabietal	−1.779434393	1.75 × 10^−04^
	Arabidiol	−1.857123891	2.21 × 10^−09^
	Copal-8-ol diphosphate	−1.924119174	1.23 × 10^−09^
	Aphidicolin	−1.947152337	3.92 × 10^−16^
	Gibberellin A53	−2.991652225	1.12 × 10^−08^
	Gibberellin A14	−3.154745064	3.50 × 10^−07^
	cis-Abienol	−3.453551139	2.83 × 10^−09^
	5alpha-Acetoxytaxa-4(20),11(12)-dien-10beta-ol	−3.88691206	6.02 × 10^−11^
	5alpha-Acetoxy-10beta,14beta-dihydroxytaxadiene	−4.134902554	7.48 × 10^−08^
Phenylpropanoid	4-Hydroxystyrene	5.642964883	7.17 × 10^−12^
	Coniferyl acetate	4.620635971	8.40 × 10^−15^
	Chavicol	3.886054817	1.89 × 10^−10^
	Caffeoyl quinic acid	3.2138012	9.27 × 10^−06^
	Scopoletin	2.505276504	1.57 × 10^−12^
	Syringin	2.135952387	6.27 × 10^−08^
	Ferulaldehyde	2.097477361	8.55 × 10^−06^
	4-Coumaroylshikimate	−1.37435324	7.53 × 10^−08^
	Coumaryl acetate	−1.832764994	1.92 × 10^−11^
	Coniferin	−2.093818248	9.62 × 10^−12^
Flavonoid	Quercetin 3-beta-D-sophoroside	2.889187046	1.49 × 10^−07^
	Glyceollin	2.745732531	3.47 × 10^−07^
	Laricitrin	2.477143429	7.15 × 10^−06^
	Quercetin 3-(2G-xylosylrutinoside)	2.476832075	2.18 × 10^−03^
	Pinobanksin 3-O-acetate	2.308305371	3.46 × 10^−13^
	Maackiain-3-O-glucosyl-6″-O-malonate	2.286077494	2.65 × 10^−04^
	2-Hydroxy-2,3-dihydrogenistein	2.247817936	4.06 × 10^−06^
	Leucoanthocyanidol	1.976381763	1.20 × 10^−04^
	Formononetin	1.870944123	1.46 × 10^−05^
	8-C-Glucosylnaringenin	1.53852511	1.01 × 10^−04^
	Glyceocarpin	1.404825044	1.19 × 10^−05^
	3′,4′,5-Trihydroxy-3,7-dimethoxyflavone	1.470845226	4.36 × 10^−06^
	Myricetin	1.145537801	5.03 × 10^−05^
	3-O-Methylquercetin	−1.216396205	1.37 × 10^−03^
	Vitexin	−1.3550808	1.35 × 10^−05^
	Rhoifolin	−1.401280595	6.82 × 10^−07^
	Malonyldaidzin	-1.699702739	3.71 × 10^−11^
	Sophoraflavonoloside	-1.720462228	5.92 × 10^−05^
	Hesperetin 7-O-glucoside	-1.779216101	1.46 × 10^−08^

**Table 2 genes-11-00187-t002:** The number of miRNAs and target genes in two cytotypes of *S. canadensis*.

Cytotype	Known miRNA Number	Target Gene Number of Known miRNA	Novel miRNA Number	Target Gene Number of Novel miRNA
Diploid	53	930	136	815
Hexaploid	54	933	135	820
Total	54	933	140	827

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
