# Peer review of "Metabolome and Transcriptome Analysis of Hexaploid Solidago canadensis Roots Reveals its Invasive Capacity Related to Polyploidy"

_genes, 2020, doi:10.3390/genes11020187_

Round 1

Reviewer 1 Report

The authors describe an omics-based study, at transcript-metabolite level, of 2 Solidago canadensis cytotypes of different ploidy level (diploid and hexaploid), in order to shed light on molecular-biochemical mechanisms related to the higher invasiveness of the hexaploid ones.

Is not not clear to me how the experimental design of metabolite data was produced. More in details:

why the authors focused, starting on the whole deleted metabolome (1,305 compounds) on terpenoids, phenylpropanoids and flavonoids, rather on all secondary metabolites. If it came by the enrichment analysis of the transcriptome data, i would swap the work-flow, firstly showing transcriptomics and then metabolomics (intended now as a "targeted" analysis based on hints coming from transcriptomics); the level of validation used for the different metabolites should be clearly indicated; this request comes by the fact i am quite afraid for a series of identifies reported in the manuscript: for instance, the authors described the presence of a series of terpenoids, especially monoterpenes, which in my experience are quite difficult to be detected by LC-MS, and are more easily detectable by GC-MS.

I also have some doubts about how the correlation networks were build: to calculate a coefficient coefficient, at least 3 values are needed: if only two experimental points are available (diploids and hexaploids), how these coefficients were generated? Were technical/biological replicates used? This is a crucial point to be clarified, which also affects the significance of the reported results.  

As general comment, in my view, it would have been much more informative to include the tetraploid genotype, together with the diploid and the hexaploid ones. This suggestion comes from findings on other Soledago species (S. gigantea), for which a higher invasiveness has been found in tetraploid cytotype (Schlaepfer et al., 08; Schlaepfer et al., 10).

Furthermore, i would also mention studies on other invasive plants (like orchids etc) for which polyploidization studies have been performed (for instance, Grosso et al., 18), or on general aspect of polyploidy-related invasiveness (for instance, Beest et al., 12).

Finally, English and text form needs some moderate revisions: for instance, line 53/68 and 64/66 (use simple present and simple past, respectively, rather than future); or lines 54-55, which should be re-phrased.

Reviewer 2 Report

Authors evaluated transcriptomic and metabolomic profiles to compare two cytotypes of S. canadensis.  They identified a number of altered metabolites, genes, and miRNAs in their analysis that are involvd in varios relevant pathways, such as biosynthesis of secondary metabolites, MAPK signaling, etc.  The paper is well organized, yet would benefit from more careful editing due to some grammatical errors.  Further, many data analysis details are lacking.

Major comments:

3, lines 103-104: Authors state “The mass ions with relative standard deviation (RSD) > 30% were removed”. Was standard deviation calculated across all samples? Were pooled QC samples included to evaluate robustness of peaks? In what order were samples run for metabolomic analyses? Further details on processing and analysis of metabolomic data should be included: What parameters were used in Progenesis for MS1 and MS2 processing? How were peaks searched against KEGG and which KEGG version was used? For differential analysis, how were p-values adjusted for multiple comparisons? FDR? In section 2.5, how were metabolites annotated? Were metabolites manually assigned a KEGG Compound ID, which was then mapped to pathways?  In section 2.5, what metabolites were used as background when performing the hypergeometric tests? How were p-values adjusted for multiple comparisons? Further details on processing and analysis of transcriptomic data should be provided: What parameters were used for Bowtie2 alignment? What versions or access date of GO, KEGG, or other databases were used? For differential analysis, how were p-values adjusted for multiple comparisons? FDR? What genes were used as the background for pathway enrichment analysis? Further details on processing and analysis of small RNA data should be provided: 4 line 179: authors should clarify what “other sRNA databases” are. Parameters for cmsearch, miRNA precursor identification, and RIPmiR should be provided. For differential analysis, how were p-values adjusted for multiple comparisons? FDR? Parameters for psRobot and TargetFinder analyses should be provided. What software was utilized to perform pathway enrichment analysis showed in Figure 2? Metabolomics data should be deposited in a public repository (e.g. MetaboLights, Metabolomics Workbench). Authors should make it clear that their metabolite identifications are putative, since they have not been verified using high quality standards. Because there is much uncertainty in identification without the use of standards, it is likely that identification errors are present.  As presented, it seems that the metabolite identification are definite, which they are not.  Also, no validation of any of the metabolites (using standards) has been reported. Regarding the metabolite-gene analysis, a discussion is lacking about specific examples that may support the validity of the network. Have some of the relationships of interest been supported by other studies?  Because the analysis is correlative in nature, it may not reflect actual regulatory processes, and may include artifacts (e.g. a given gene and metabolite may be correlated, without their being any biological connection).  Authors should discuss this caveat. Regarding the metabolite-gene network: how is the importance of a node evaluated? Figure 8: the labels are too small and not legible. Figures S3 and S4 should include statistical significance. Authors should consider removing the word “Integrated” from the title, since no follow-up work was performed to validate associations.

Minor Comments:

Some citations are missing (e.g. Blast2GO, MapMan, KEGG, GO, etc.) For Table S1: it is unclear what the “P-value” and “VIP” columns refer to. For Table S2: an added column providing additional metabolite classification (e.g. terpenoids, phenylpropanoids, flavonoids) would be helpful. For Table S3: added columns providing associated GO terms would be helpful. Table 1: the location of the horizontal lines is unclear: do they provide boundaries for the 3 types of classifications? 8, line 224: authors mention “all metabolites”, but presumably they mean all differentially abundance metabolites that map to pathways? Figure S2: it may be helpful to sort the barplot by increasing pathway size. Figure 3: legend is not legible and should have a title. Figures 4, 5, and 6: legend needs a title. Describe the acronym “DEG” in the abstract. Regarding the metabolite-gene network: how is the importance of a node evaluated? Figure 8: the labels are too small and not legible. Figures S3 and S4 should include statistical significance.

Round 2

Reviewer 1 Report

The authors have addressed all my previous comments.

Reviewer 2 Report

The following 3 points must be addressed:

Point 2: what metabolites were used as background when performing the hypergeometric tests? How were p-values adjusted for multiple comparisons?

Response 2: Thank you for your questions. The background was the total putative metabolites. The p-values were adjusted by FDR.

With this response, it is still unclear what “putative metabolites” entails.  Do authors mean that all putative identified metabolites in their experiment was used as background?  This should be clarified in the text.

Point 6: Metabolomics data should be deposited in a public repository (e.g. MetaboLights, Metabolomics Workbench).

Response 6: Thank you for your suggstions. According to your suggestion, the metabolomics data has been submitted to the Metabolomics Workbench database.

Authors must provide a project ID and link so that readers can readily access the deposited data.

Point 11: Some citations are missing (e.g. Blast2GO, MapMan, KEGG, GO, etc.). For Table S1: it is unclear what the “P-value” and “VIP” columns refer to.

Response 11: Thank you for your suggstions. The website of Blast2GO, MapMan, KEGG and GO etc. were provided in revised manuscript. The p-value and VIP was used for differentially accumulated metabolite analysis. There is no need in Table S1, so we have removed it from this revision. These two values were introduced in the section 2.4.

Not all citations have been added in.  One example, which is not the only one is Blast2GO (4 citations ar here https://www.blast2go.com/support/testimonials)
